# Gender Differences in Emotional Valence and Social Media Content Engagement Behaviors in Pandemic Diaries: An Analysis Based on Microblog Texts

**DOI:** 10.3390/bs13010034

**Published:** 2022-12-30

**Authors:** Ran Feng, Alex Ivanov

**Affiliations:** School of Media and Communication, Shanghai Jiao Tong University, No. 800 Dongchuan Road, Minhang District, Shanghai 200240, China

**Keywords:** pandemic diaries, gender differences, emotional valence, content engagement, pandemic proximity

## Abstract

The effects of the COVID-19 pandemic are individualized, which means that our emotions and behaviors would experience changes of different degrees. These changes have led to subtle connections within the social media context. This study concentrates on pandemic diaries posted on microblog sites during the lockdown period in China and explores the association between gender, emotional valence in diaries, and social media content engagement behaviors. Through computational methods, this study found that males and females tended to present significantly different emotional valence and social media content engagement behaviors. A negative correlation existed between emotional valence and comment behavior in female diary texts. Moreover, the pandemic proximity had a moderating effect on emotional valence and social media content engagement behaviors. This article attempts to explain the emotional and behavioral characteristics related to social media diaries and express concerns for the emotional health of disadvantaged blog users in the severely affected area during the pandemic.

## 1. Introduction

In 2020, a sudden outbreak of COVID-19 in China rapidly became a public health crisis affecting the country and the world as a whole. Living during a severe pandemic, people get used to posting texts, pictures, and videos on social media to record daily moments and express feelings, and actively participate in interactive behaviors, e.g., posting, retweeting, commenting, and clicking thumb-ups. Emotional contagion on social media leads people to experience the same positive or negative emotions without their awareness [1]. This influence varies from person to person and could change the emotional state and interactive behaviors. Within this process, what kind of role does gender play? What kind of emotional tendency do people’s information-related behaviors present in general? How do the factors mentioned above affect interactive behaviors on social media? Starting with these questions, the study examines the online diary texts posted by Sina Weibo users during the first lockdown period in China. The authors pay attention to the relationship between gender, emotional tendencies in the online diaries, and social media content engagement behaviors, and compare the situation in the severely affected area with others.

## 2. Literature Review

### 2.1. Gender Differences in Emotional Experience during the Pandemic

Emotional valence is an important concept, which generally refers to the experience of pleasant–unpleasant (positive–negative) feelings [2], reflecting people’s judgment on the value of information [3]. The pandemic impacts the emotional health of males and females differently. Although fewer women than men die from COVID [4], the employment situation and family life of women have suffered more in relation to men. For one, women constitute the majority of healthcare workers, and they bear the brunt of the virus in emergency rooms [5]. Women are also more vulnerable to losing their jobs due to layoffs or other reasons [6,7]. During quarantine and lockdown periods, women are the ones who take care of the increased chores in the household and care for family members [8]. Even the risk of domestic violence has risen as a result [7]. The increased workload at home and the poorer employment opportunities significantly worsen women’s emotional anxiety during the pandemic [9].

Meanwhile, men and women differ in their crisis emotion coping strategies [10,11]. Women were found to use a greater variety of coping strategies [12], and men possess higher self-efficacy due to better emotion regulation [7]. On the other hand, although women may be at a disadvantage in terms of education and social expectations, their response to danger is emotionally stronger than men [13]. Therefore, the authors propose the first hypothesis:

**H1.** *There are significant gender differences in terms of emotional valence in social media pandemic diaries*.

### 2.2. Gender Differences in Social Media Content Engagement Behaviors

The gender difference in response to risk may have a particular impact on online interactive behavior strategies on social media. One study found that men usually show a self-centered style when facing crises, whereas women are more aware of the overall situation [14]. Since women are generally more empathic and collaborative, they may be more inclined to come out with more diversified beliefs and the potential to reach a consensus [14]. Women often try to understand their relationships with their family and friends through social activities, thinking about self-worthiness, and seeking change [15]. During the lockdown stage, the online female community helped desperate people by delivering pleas for assistance and actively launching donation activities [4].

Besides, there are gender differences in communication styles as well [16]. Females prioritize supportive communication, while men focus on problem-focused conversation [17,18]. Women value close relationships for their emotional and expressive qualities, whereas men tend to conceptualize close relationships for their instrumental features [17]. The authors suggest that these gender differences would be reflected in the frequency of online social media content engagement behaviors, e.g., commenting, retweeting, and clicking thumb-ups. Hence, the second hypothesis is proposed:

**H2.** *There are significant gender differences in the frequency of commenting, retweeting, and clicking thumb-ups of social media pandemic diaries*.

### 2.3. Emotional Valence and Social Media Content Engagement Behaviors

Emotions are expressed externally on social media platforms, spawning information dissemination behaviors [19]. Research on Facebook and Twitter has demonstrated that the sentiment of messages could trigger more cognitive involvement [20], which was correlated with retweeting, commenting, and liking [1,21]. People prefer retweeting posts that correspond to the emotional valence of the respective event [22]. User posts with prosocial emotions can trigger retweeting behavior [23]. Researchers found that commenting generally occurs more commonly with exposure to negative emotions [24]. Blog users prefer clicking likes for messages which convey emotions that correspond to the base emotion of an event. For example, a previous study found that cancer-related tweets with joy and hope received more likes [25].

The heuristic–systematic model suggests that individuals making a judgment about health issues tend to process information either heuristically or systematically [26]. If a benign situation elicits positive emotions, we are likely to perform heuristic processing, but if a problematic situation elicits negative emotions, we tend to deal with it more systematically [18]. Social media content engagement behaviors, such as retweeting and clicking likes, are content-integration behaviors, which reflect the user’s recognition of someone else’s opinions. They are cooperative behaviors more affected by positive emotions [19]. On the contrary, comments and postings would be more affected by negative emotions and more likely to be associated with content creation. With conflicts of opinion, users who write comments or post original ideas would think objectively to express their attitudes and opinions [19]. Based on the literature above, the authors pose the following hypothesis:

**H3.** *The emotional valence of social media pandemic diaries is significantly associated with the frequency of commenting, retweeting, and clicking thumb-ups*.

### 2.4. The Moderating Effect of Pandemic Proximity

The spatial association is significant for health crisis perception. Pandemic proximity refers to the proximity to the pandemic in terms of the geographical environment, community, and infected populations [27]. A previous study found that the “distance proximity effect”, signifying an individual’s physical or psychological distance to an epicenter, affects their perception of the pandemic [28]. Some studies also discovered that in areas with more stern outbreaks, the public was more prone to serious psychological stress, anxiety, and related behaviors [29,30]. During the outbreak of the pandemic, cases of COVID-19 came mainly from Hubei Province, which became the province with the highest number of infections and deaths [31]. The attention of the whole country was attracted to Hubei province, while its medical resources were most limited. Compared with other Chinese cities, residents in Hubei were closer to infected people in their community or infected relatives, friends, or acquaintances around them. Thus, this study would further explore the research question:

RQ: How does the pandemic proximity moderate the relationship between gender, emotional valence, and the frequency of commenting, retweeting, and clicking thumb-ups of social media pandemic diaries?

## 3. Methods

### 3.1. Data Collection

Social media platforms are known as important channels for obtaining and evaluating public opinion on crisis events [31]. Ever since the official announcement of the Wuhan Lockdown in late January 2020, the pandemic diaries on microblogs have gradually accumulated large, complex, and diverse data, which constitutes a rich corpus for analysis. Sina Weibo (also known as Weibo) is the most popular microblog site in China [32]. Weibo had over 500 million monthly active users and 200 million daily active users in 2020. It was selected to be the social media platform from which we collected the diary texts. Users can access Weibo through various mobile devices, such as personal computers and mobile phones, and realize instant information sharing, communication, and interaction in texts, pictures, and videos [33].

In this study, the authors compiled Python computational language to collect pandemic-related diary posts on microblogs. The representative topic words for searching included “quarantine diary”, “Wuhan diary”, “pandemic diary”, and “lockdown diary”. The types of diary posts were mainly plain texts, texts with images, and texts with videos. The period for data collection was from 0:00 a.m. on 23 January 2020, to midnight on 8 April 2020 (77 days). This overlaps with the Wuhan lockdown period. During this time, the overall pandemic situation in Hubei was more severe than that outside the province in terms of confirmed cases and calamities [34]. The current study extracted the following information: user ID, gender, place of residence, the number of followers, timestamp (post time of messages), diary posts, the number of comments, retweets, and thumb-ups. These data are open sources on Weibo and can be easily found on the user’s homepage. Researchers deleted duplicate and unnecessary data (the number of pictures and videos affiliated with each post, as well as user certification level) and finally accumulated 46,439 valid diary samples.

### 3.2. Data Analysis Procedure

The emotional valence of microblog texts is based on the sentiment analysis of the machine learning method since it is more accurate than the semantic dictionaries and can be used in diverse scenarios. This study used the Natural Language Toolkit (NLTK) in Python package to distinguish emotions and then extracted a model to classify the text data to calculate the final tendency probability. The overall sentiment index of each text was first determined, and the sentiment was classified into positive and negative categories. In the full algorithm for binomial classification of the texts, the model applied samples pre-coded with two types of labels as the training set and then used a Bayesian formula to calculate the attribution and probability of the pending texts. The researchers manually coded a random 10% of the total collected text, assigning each text a positive or negative attribute. This portion of the text then became the training set for machine learning to construct the classifier. After processing each post, the researchers obtained a sentiment index with a value between 0 and 1. The smaller the index (closer to 0), the more negative the sentiment, and vice versa. Scores smaller than 0.45, between 0.45 and 0.55, and bigger than 0.55 were viewed as negative, neutral, and positive emotions, respectively. After calculating the sentiment index of all samples, we imported the data into SPSS 24.0 for analysis and hypotheses testing as well as answering the research question. In line with existent literature, this study defines the frequency of social media content engagement behaviors as how often users interact with other individuals or groups through symbolic exchanges of information and emotion via retweeting, clicking thumb-ups, and commenting on posts [35].

## 4. Results

### 4.1. Descriptive Results

In a total sample of 46,439 posts, 13,802 (29.7%) posts were from male users, 32,637 (70.3%) were from female users; 24,463 (52.7%) were posted by users in Hubei Province, and 21,976 (47.3%) were posted by users outside Hubei. Most users have between 100 and 400 followers. A minority of users have over 10,000 followers. The number of posts containing texts with images is the largest (24,634), followed by plain text posts (15,635), while posts with both text and video are the least (6170). Details are shown in Table 1.

According to the White Paper of the Chinese People’s Government, this paper extracted and analyzed the Weibo data sets in three stages of COVID-19. The first stage (from 23 January to 20 February) refers to the breaking out and spread of the pandemic. The second stage (from 21 February to 17 March) stands for the declining trend of confirmed cases and situations under control. The third stage (from 18 March to 8 April) represents the recovery and return to normality [34]. Generally speaking, during the pandemic, the emotional index of the blog post samples tended to be positive (Figure 1). More than half of the blog post indices were between 0.9 and 1, reflecting quite a positive emotional tendency. There were a few blog posts with negative emotions. Only 14.30% of posts presented a very negative emotional state, with emotional valence indexes below 0.1. The distribution of blog posts’ sentiment index has a slightly polarizing trend. The fluctuation of the average emotional index of males and females was not always consistent (Figure 2).

### 4.2. Hypotheses Testing Results

This study’s hypotheses test the relationship between gender, emotional valence, information interaction behavior, and pandemic proximity. To start with, we want to find whether there are significant differences in the emotional valence of microblog pandemic diaries posted by users of different genders. The independent sample *t*-test result shows that the sentiment index of diary blog posts from men and women is significantly different (t_(46437)_ = 4.273, *p* < 0.001). The sentiment index of women’s diary posts (M = 0.748, SD = 0.363) is significantly higher than that of men’s (M = 0.732, SD = 0.375). The difference in the average sentiment index is 0.016, 95%CI = [0.009, 0.023]. Therefore, H1 is supported.

Next, the authors examined whether there is a significant difference in the frequency of social media content engagement behaviors (clicking thumb-ups, retweeting, and commenting) of male and female users. Table 2 presents the mean and difference in frequency of three interactive behaviors between male and female blog posts. The results of the independent sample *t*-test show that there is a significant gender difference in the number of “likings” (t_(46437)_ = −2.877, *p* < 0.01). The number of thumb-ups in males’ posts is significantly higher than that of females. There is a significant gender difference in the reposting diaries (t_(46437)_ = −2.365, *p* < 0.05). The retweeting volume of male blog posts is significantly higher than that of females. The posts of the two genders also showed significant differences in the number of comments (t_(46437)_ = −3.011, *p* < 0.01). The number of comments on males’ posts is significantly higher than that of females. H2 is thus supported.

Regarding the correlation between the emotional valence of the diary text and three types of social media content engagement behaviors, contrary to our expectations, these results are not significant. However, we find the emotional valence of females’ blog posts is significantly negatively correlated with the frequency of comment behavior (r = −0.01, *p* < 0.05). Females’ emotional valence is not significantly correlated with the behavior of clicking likes and retweeting. There is no significant correlation between the emotional index of males’ blog posts and social media content engagement behaviors. Therefore, H3 is not supported.

Then, the authors examined the potential moderating effect of pandemic proximity on the relationship between gender and emotional valence and the relationship between gender and social media content engagement behaviors. The authors used Macro PROCESS in SPSS for data analysis. Categorical variables were transformed as dummy variables (0 for women and 1 for men; 0 for non-Hubei provinces and 1 for Hubei province) and then the moderating effect was tested. Using 5000 bootstrap samples and a 95%CI, Model 1 was used to address RQ [36]. The analysis results show that the main effect of the model of gender differences affecting the emotional valence is significant (F_(3, 46435)_ = 357.18, *p* < 0.05), as is the interaction between gender and pandemic proximity (F_(1, 46435)_ = 390.61, *p* < 0.05). The main effect of gender difference affecting comment behavior is also significant (F_(3, 46435)_ = 6.16, *p* < 0.05), as well as the interaction term (F_(1, 46435)_ = 5.23, *p* < 0.05). The main effect of gender difference affects giving thumb-ups is significant (F_(3, 46435)_ = 5.80, *p* < 0.001), too, as is the interaction term (F_(1, 46435)_ = 4.37, *p* < 0.05). Finally, while the main effect of gender differences affecting retweeting behavior is significant (F_(3, 46435)_ = 3.64, *p* < 0.05), the interaction term is not so (F_(1, 46435)_ = 2.81 *p* = 0.09). The moderating effect of the pandemic proximity could be viewed in Table 3.

## 5. Discussion and Conclusions

This study focuses on pandemic diaries posted on Weibo during the lockdown period in China and explores relationships between gender, emotional valence, and social media content engagement behaviors with diary texts. We found that males and females had significantly different emotional valence and social media content engagement behaviors. A negative correlation existed between emotional valence and commenting behavior in female diaries. The pandemic proximity had a moderating effect on emotional valence, commenting, and clicking thumb-ups for blogs.

Previous experience with the infectious epidemic has demonstrated that incorporating gender analysis into the intervention of the epidemic can help promote gender equality and achieve the goal of health equity [37]. The emotion change tendency always showed an abrupt and statistically significant sentiment decline around the beginning of the pandemic, followed by an asymmetric and slower recovery [38]. A good grasp of the characteristics of male and female media users’ mood fluctuations and online interactive behaviors is of great significance for mass communication research in the post-pandemic era.

In terms of the relationship between gender and the emotional valence of diary texts, the study finds that both men and women demonstrate positive emotions, but women showed a significantly higher degree of emotional positivity than men. Although women presented more negative emotions than men at the beginning of the epidemic, as the domestic situation tended to improve, women’s negative emotions gradually declined [7]. Our findings also suggest that women have some inherent characteristics and resources, such as self-esteem and unique personal experience, which are necessary for effective coping with stress [39]. The finding reminds us to pay attention to women’s self-reflection and caring abilities during a crisis. The female bloggers in our study may have good adaptability, reflection, and motivation, so they presented more positive emotions in the diary.

As for the relationship between gender and the information interaction behavior of the diary text, we find that the frequency of clicking likes, reposts, and comments in men’s diaries is significantly higher than that of women. The findings of the study indicate that men’s blog posts have received more attention and are more likely to stimulate interactive behavior. It is common to find gender disparities in the decision to go online, the perception of the technology, the type of online activity participated in, and the frequency and motivation in engaging in online activities [16], but the authors still have to look into the gender inequality implication in social media behaviors. Judging from the pandemic diary texts on social media, although women have more opportunities to speak out than before, men still dominate the discourse in the online world.

When it comes to the association between emotional valence and information interaction behavior, on the whole, the emotional valence of the pandemic diary is not significantly related to the interaction behavior, and this finding does not conform to prior predictions based on the heuristic–systematic model. One possible explanation is those who react to the positive posts would step up to a higher level of news engagement to share and comment [40], but most of the microblog users in our study did not engage deeply, probably because of the tiredness induced by information overload during the pandemic. However, we find that particularly in women’s diary posts, the emotional valence and comment behavior show a negative correlation. The more emotionally positive the diary was, the fewer comments it had, and vice versa: the more emotionally negative the diary, the more comments it contained. Many comforting and encouraging comments existed below women’s diaries of negative emotion, expressing hope, support, and encouragement to bloggers in low spirits. This reflects the characteristics of women emphasizing partnership and empathy.

Concerning the role of pandemic proximity in the above-mentioned relationship, the study discovered that it has a moderating effect on the relationship between gender and emotional valence, the relationship between gender and clicking likes, and the relationship between gender and commenting behavior. The gender difference in the emotional valence of Hubei and non-Hubei diary posts shows opposite trends. Female bloggers in Hubei and male bloggers in locations outside Hubei are presented with more negative emotions than their counterparts. The gender differences in the number of comments and thumb-ups are greater in non-Hubei diary posts, but not in Hubei pandemic diaries. The closer the bloggers are to the center of the pandemic, the smaller the gender difference in the frequency of commenting and clicking likes would be. The geographic dimension is positively related to both liking and commenting behaviors. These findings demonstrate that pandemic proximity is an important variable associated with people’s emotions and behaviors during the pandemic, and future studies need to further consider the influence of regional factors on these relationships. It is encouraged that more scholarly attention should be paid to the emotional health of people who are continuously involved in lockdown and quarantine.

Admittedly, this work has two limitations. In terms of data collection and analysis, the data in this study were not comprehensive enough. Some personal attributes were hidden by privacy settings and could not be included in the regression analysis. Secondly, the study only selected the microblog diary posts during the first pandemic lockdown period in China for analysis. Considering the long-term coexistence of the epidemic with human society, it is necessary to perform longitudinal research in the future to explore potential long-term differences and associations.

The major contribution of this study lies in the discovery of the differences in emotional valence and social media content engagement behaviors of pandemic diaries posted by users of different genders. Against the backdrop of the pandemic, the global health issue is also a crisis of information [41]. At this time, researchers need to examine various factors that affect social media emotions and discuss the possible psychological consequences. We hope that this research will help the public improve their understanding of the emotional communication and behavioral mechanisms of social media during the COVID-19 pandemic.

## Figures and Tables

**Figure 1 behavsci-13-00034-f001:**
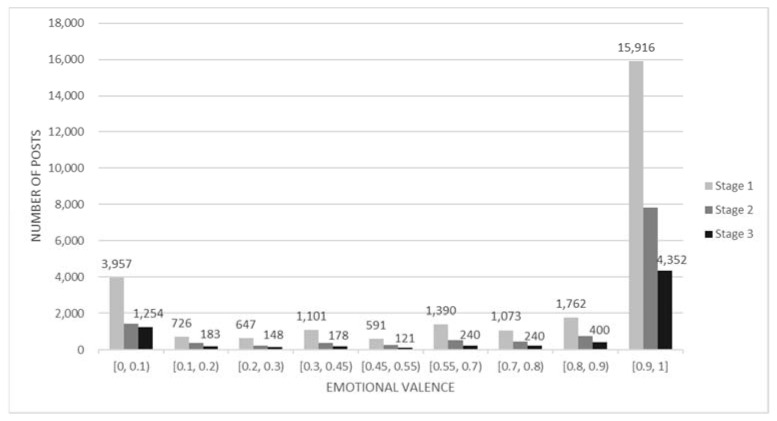
Segmented statistical graph of the emotional valence index of diary posts.

**Figure 2 behavsci-13-00034-f002:**
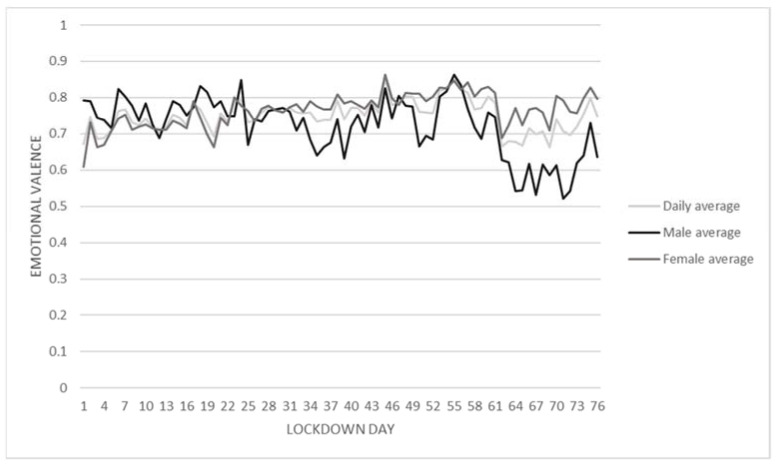
The overall trend of average emotional valence in the Wuhan lockdown.

**Table 1 behavsci-13-00034-t001:** Descriptive statistics of Microblog users and diary texts.

	Followers	Following	Thumb-Ups	Retweets	Comments
Mean	139,552	562	64	7	10
Median	476	332	2	0	0
Mode	1	1298	0	0	0
Quartile	150	157	0	0	0
SD	1,469,106.33	964.99	2877.30	665.39	219.81

**Table 2 behavsci-13-00034-t002:** Statistical differences in social media content engagement behaviors.

Information Interaction Behavior	Gender	N	Mean	SD	Mean Difference	95%CI
Clicking likes	Male	13,802	123.51	4854.81	−84.05	[−141.31, −26.80]
	Female	32,637	39.45	1345.74		
Retweeting	Male	13,802	18.68	1215.87	−15.98	[−29.22, −2.73]
	Female	32,637	2.71	68.92		
Commenting	Male	13,802	14.49	359.88	−6.72	[−11.09, −2.35]
	Female	32,637	7.77	118.19		

**Table 3 behavsci-13-00034-t003:** The moderating effect of pandemic proximity (PP).

	B	SE	t	95%CI
Gender→emotional valence				
Gender	−0.07	0.005	−13.74	[−0.08, −0.06]
Gender × PP	0.15	0.008	19.76	[0.13, 0.16]
Gender→clicking likes				
Gender	131.49	40.218	3.27	[52.66, 210.32]
Gender × PP	−123.84	59.263	−2.09	[−230.99, −7.68]
Gender→commenting				
Gender	10.75	3.072	3.50	[4.73, 16.77]
Gender × PP	−10.25	4.527	−2.26	[−19.13, −1.38]

## Data Availability

The most important data presented in the study are included in the article, further inquiries about the original data can be directed to the corresponding author.

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
