# Peer review of "Gender Differences in Emotional Valence and Social Media Content Engagement Behaviors in Pandemic Diaries: An Analysis Based on Microblog Texts"

_behavsci, 2022, doi:10.3390/bs13010034_

Round 1

Reviewer 1 Report

The article is interesting and worth considering. However, the author needs to expand the Literature review to better establish emotional valence and the social media interaction of the social media users. This can also be achieved if some form of qualitative content analysis was done from a sample size of users. 

Author Response

Many thanks for the comment and suggestion. Our responses are as follows: (1) We have expanded the literature review, especially in the section of "2.3 Emotional Valence and Social Media Content Engagement Behaviors" in order to establish a potential association between emotional valence and social media interaction. Some studies on Facebook, Twitter, and other social media platform are included, and we found that most of them are consistent with the Heuristic-Systematic Model. And we also compared our results with previous studies in the discussion.  (2) Frankly speaking, we didn't do a qualitative content analysis because it is hard to find an adequate coding frame for this study. Since this is an explorative work, we believe that the next step is to develop a rigorous system to deal with the posts. (3) We have done the necessary language polish. Some grammatical mistakes have been corrected. 

Reviewer 2 Report

Title: Gender Differences in Emotional Valence and Information Interaction Behaviors in Pandemic Diaries: An Analysis Based on 3 Microblog Texts

Comments:

1.       The succinct discussion of the paper is truly commendable. In pithy words, the juxtaposing of concepts – gender, emotional valence, informative interaction behavior and pandemic social distance – was admirably executed in a well-woven narrative.

2.       The article is uniquely presented  - fresh ideas, an interesting apposition of variables and creative interlacing of narratives. The strength of the article is perceptibly displayed in its contribution to scholarship, the knowledge gap identified, the cohesiveness of thoughts, the methodological accuracy, the complete and delicate referencing and the airtight presentation of arguments.

3.       Scientific content is empirically sound and prudently tested. Interpretation of the data coheres with the statistical analysis, expressing pithily the significant correlation between gender differences and emotional valence in diaries, gender differences and frequency of commenting, retweeting and thumb-ups of social media diaries, and the vital findings on the correlation between emotional valence and information interaction behaviors. The current findings are even combined with previous findings (references supplied by the writers) on the possible reasons for the correlation.

4.       The writing is a mix of simplicity and profundity in writing style and the ease of understanding it offers for the general readership of the article.

5.       The conclusion answers the research questions stated and the hypotheses, soundly tested and measured, were carefully supported or refuted with corresponding statistical analysis of the data.

6.       Observed findings, though with stated limitations, were ingeniously and adequately extrapolated, transcending whatever barriers they encountered in obtaining research data.

7.       Minor grammar lapses were observed particularly in lines 19 and 123.

Author Response

Many thanks for your suggestions and approval. Our responses are as follows: (1) To enhance the rigor of the present study, we further modified two important theoretical concepts: information interaction behavior and pandemic social distance. According to the previous literature, the term "social media content engagement behaviors" is more accurate to describe interactive behaviors like commenting and liking, while "pandemic proximity" would be better to replace the original geographical dimension. (2) We have corrected the grammatical mistakes in line 19 (the last sentence of the abstract) and line 123 (the first paragraph of data collection). And we also made several language polishes throughout the article. We hope that these modifications would improve the overall quality.

Reviewer 3 Report

This work uses an innovative method and statistical analysis that can enhance knowledge in the area but, from my perspective, it currently has several limitations:

Regarding the structure, I believe that the manuscript would benefit from a revision. I would recommend deleting the information included in the second paragraph of the introduction because it repeats the results already presented in the abstract. I suggest keeping the introduction title, revising and improving the reasoning of the first paragraph, and moving on to the information in the “literature review” section without inserting a new title. I also suggest moving the hypotheses to the end of the introduction, after the presentation of the study’s objectives.

Overall, figures need to be improved to facilitate their understanding.

I also recommend ending the article with “Major contributions” instead of “Limitations”

From my perspective, both the literature review and the overall manuscript need to have a more in-depth framing, including the support of the statements with different references. Many of these claims need to be substantiated by further research.

Additionally, the manuscript still needs a review in order to clarify some statements, for example:

28-29 “Social media facilitate the transfer of emotion, leading people to experience the same feelings without their awareness”

59: Women have more of these strategies at their disposal

102 Pandemic social distance refers to the proximity to the pandemic in terms of geographic environment and interpersonal relationships [22] ---- Social distance has a completely different definition from the one used by the authors. Is the definition used well established? I was not able to find paper 22 included as the reference.

140 sentiment analysis of the machine learning methods and Toolkit to classify emotions

Please explain the relationship between emotions and feelings.

Provide a reference to the toolkit presented.

The “microblog texts” have privacy restrictions? I strongly recommend including information regarding ethical considerations in the methods section, in addition to the limitations section.

Minor suggestions:

35 avoid the use of “we” ---- The authors…

36 Clarify “emotional tendencies”: What does this mean?

information interaction behaviors --- Please clarify this concept and be consistent. Is this the same as Online Interaction Behaviors (line 66)?

Please conduct a proof-reading throughout all the manuscript to detect minor typos, for unnecessary data and finally accumulated 46,439 137 ---- please provide examples of “unnecessary data”

example:

60 -61 The study assumes that male and female bloggers would present to have different emotional valence in their diaries
